# Making Large Language Models Perform Better in Knowledge Graph Completion

## ABSTRACT

Large language model (LLM) based knowledge graph completion (KGC) aims to predict the missing triples in the KGs with LLMs. However, research about LLM-based KGC fails to sufficiently harness LLMs' inference proficiencies, overlooking critical structural information integral to KGs. In this paper, we explore methods to incorporate structural information into the LLMs, with the overarching goal of facilitating structure-aware reasoning. We first discuss on the existing LLM paradigms like in-context learning and instruction tuning, proposing basic structural information injection approaches. Then we propose a **K**nowledge **P**refix **A**dapter (KoPA) to fulfill this stated goal. The KoPA uses a structural pretraining phase to comprehend the intricate entities and relations within KGs, representing them as structural embeddings. Then KoPA communicates such **cross-modal structural information understanding to the LLMs** through a knowledge prefix adapter which projects the structural embeddings into the textual space and obtains virtual knowledge tokens positioned as a prefix of the input prompt. We conduct comprehensive experiments and provide incisive analysis concerning how the introduction of cross-modal structural information would be better for LLM's factual knowledge reasoning ability. Our code and data are available at anonymous.4open.science/r/KoPA-3415.

## CCS CONCEPTS

• **Information systems** → *Information integration*; • **Computing methodologies** → *Natural language generation*; *Semantic networks*.

## KEYWORDS

Knowledge Graphs, Knowledge Graph Completion, Large Language Models, Graph-text Fusion, Cross-modal Adapter

## 1 INTRODUCTION

Knowledge graphs (KGs) [2] are the quintessential wisdom essence and key infrastructure of modern AI. KGs represent and store real-world knowledge in the triple form: (*head entity, relation, tail entity*). This structured format of knowledge triples offers significant advantages across many AI fields such as recommendation systems [30], question answering [42], and fault analysis [7]. However, there is a pertinent drawback of KGs, whether manually curated or automatically extracted. Their scope is restricted to observed knowledge,

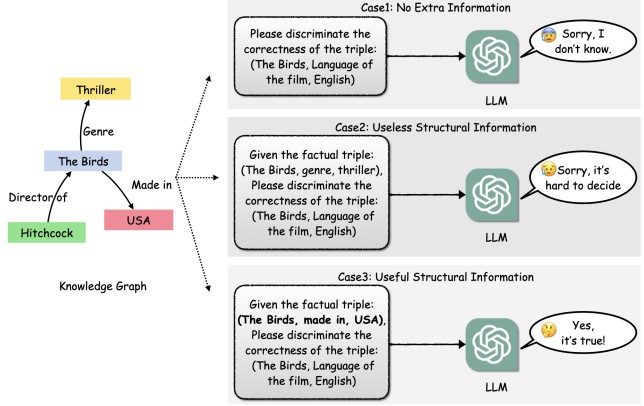

**Figure 1: A simple case of LLM-based KGC. Useful structural information that describes the surrounding information about the entities can serve as auxiliary prompts and guide the LLM to make correct decisions.**

resulting in an **incomplete** representation riddled with unobserved or missing triples. This phenomenon motivates knowledge graph completion (KGC), which aims to predict the missing triples and further enhance the given KG.

Existing KGC approaches can be divided into two categories: methods based on embeddings [3] and pre-train language models (PLM) [40]. Recently, as large language models (LLMs) [23, 46] show outperforming capabilities [24], this field has recently been revolutionized by LLMs. Some works [41] make the first step towards LLM-based KGC, employing existing paradigms like zero-shot reasoning (ZSR) [4] and instruction tuning (IT) [24] to accomplish the KGC task. However, such approaches transform the KGC task into a text-based prediction of individual triples, leading to specific fundamental problems. LLMs lack the depth and precision of factual knowledge which always results in the hallucination [48] problem of LLMs. Besides, the structural intricacies of KGs such as subgraph structure, relational patterns, and relative entities/relations are often overlooked. This richly **non-textual structured information**, if properly incorporated, can significantly enhance the LLM's understanding and representation of KGs. Figure 1 presents an intuitive view of the importance of structural information for LLM reasoning. However, this is neglected by vanilla ZSR and IT approaches [41] because each input typically only includes a single input triple, leading to potential wastage of the structural information inherent in the KG. Such an approach fails to equip the LLMs with the awareness of the KG structure.

To address these issues, we take a strategic step to LLM-based KGC, aiming to explore how to incorporate the KG structural information into the LLMs and enable structure-aware reasoning. Our initial focus involves transferring the existing LLM paradigms such as in-context learning (ICL) [9] and instruction tuning (IT) [24] to a

structure-aware context. We propose a structure-aware ICL method and a structure-aware IT method as the base models, focusing on integrating the KG structural information into LLM through text form. Such an approach benefits from the fact that specific textual information exists about entities and relationships in KG so that we can use the text to represent this knowledge as complementary background information, expecting that LLMs can learn the local structural information in KG through textual prompts. But they also have the obvious disadvantage that there is a clear semantic divide between structural and textual information. The textual descriptions in the expanded prompt still fail to fully exploit the structural information in the complex KG.

Additionally, we propose a novel **Kno**wledge **P**refix **A**dapter (KoPA) approach to make LLMs a better knowledge reasoner, leveraging **structural embedding pre-training to capture the KG structural information**. Then KoPA transforms the structural embeddings into textual embedding space by a knowledge prefix adapter and obtains several virtual knowledge tokens. These tokens, acting as **prefixes in the input prompt sequence**, direct the instruction-tuning process, providing valuable supplementary input triple information. This mapping of structural embeddings to textual form provides auxiliary information to input triples. Besides, we conduct comprehensive analysis and experiments, highlighting the remarkable performance and transferability of KoPA. In summary, our contribution is three-folded:

- **Extending the existing LLM paradigms.** We are the first extensive investigation of LLM-based KGC methods, specifically by incorporating KG structural information to enhance the reasoning ability of LLMs. We discuss the pipeline to adapt the existing LLM paradigms like ICL and IT to a structure-aware setting for KGC using addtional textual prompts.
- **Designing new cross-modal LLM paradigm.** We further propose a knowledge prefix adapter (KoPA) that effectively integrates pre-trained KG structural embeddings with LLMs. KoPA fosters comprehensive **cross-modal interactions** between textual embeddings from LLMs and structural embeddings sourced from KGs to enhance LLM's reasoning ability.
- **Comprehensive evaluation.** We conduct extensive experiments on three public benchmarks and evaluate the KGC performance of all the structure-aware methods proposed by us with adequate baseline comparison with further exploration of the transfer ability and knowledge retention degree.

## 2 RELATED WORKS

### 2.1 Knowledge Graph Completion

Knowledge graph completion (KGC) [37] is an important topic in the KG community, aiming to mine unobserved triples in a given KG. KGC contains several sub-tasks such as triple classification [3], entity prediction [3]. The common point among KGC tasks is to establish an effective mechanism to measure the plausibility of the triples. The mainstream KGC methods can be divided into two categories: embedding-based and PLM-based methods. Embedding-based methods [3, 31, 34, 38] are designed to embed the entities and relations of KGs into continuous representation spaces. These approaches make full use of structural information from the KGs to model triple plausibility with a well-designed score function and

learn the entity/relation embeddings in a self-supervised manner. Moreover, PLM-based methods consider KGC as text-based tasks by fine-tuning pre-trained language models [8]. The short textual descriptions are organized as an input sequence and encoded by the PLMs. KG-BERT [40] is the first PLM-based method that models KGC as a binary text classification task. Subsequent works like MTL-KGC [16] and StAR [35] have further improved KG-BERT by introducing more training tasks such as relation classification and triple ranking and more complex triple encoding strategy. PKGC [21] utilizes manual prompt templates to capture the triple semantic. Other methods like KGT5 [5, 29] make a step on the generative KGC [43] in a sequence-to-sequence paradigm with encoder-decoder PLMs like T5 [27]. PLM-based methods leverage the power of PLM but make the training process into text-based learning, which is difficult to capture complex structure information in the KGs.

### 2.2 LLMs for KG research

In recent years, large language models (LLMs) [23, 33, 46] have made rapid progress and demonstrated powerful capabilities in a considerable number of text-related tasks [49]. LLMs are usually pre-trained in an auto-regressive manner with next word prediction task [4] and demonstrate strong capability on text comprehension and generation. Among the research topics of LLM, integrating LLM and KG [25] is a popular and important one. On the one hand, hallucination [39, 48] is widespread in LLMs which means LLMs are lack factual knowledge and not interpretable. KGs that store structured knowledge can mitigate such a phenomenon [10, 15, 26] by introducing factual knowledge into LLMs. On the other hand, LLMs can benefit KG-related tasks such as KGC [51, 52], entity alignment [47], and KGQA [1] by its powerful generation capability. KGs for LLMs (KG4LLM) and LLMs for KGs (LLM4KG) are both important research topics. We focus on applying LLMs in the KGC task (LLM4KGC), which has not been carefully studied yet. KGLLaMA [41] made the first step by vanilla instruction tuning approach but it lacks in-depth and systematic exploration about how to unleash the power of KGs themselves to make structure-aware reasoning in LLMs and achieve better KGC performance. In this paper, we will dive into this problem from a more systematic perspective with the knowledge graph completion task.

### 2.3 Incorporate Non-textual Modality Information into LLMs

As LLMs demonstrate generalizable capabilities on text generation, many other works attempt to incorporate non-textual modality such as images [19, 50], audio [22], and video [22], which are also called multi-modal LLMs [45]. These methods tend to encode non-textual information through the modality encoders and then process it as virtual text tokens. The non-textual tokens are aligned with the word tokens by instruction tuning on multi-modal datasets.

The multi-modal LLM mentioned above usually excludes graph, which is another important data modality. There are also some works talking about how to incorporate graph data into LLMs. Drug-Chat [17] proposes to encode the drug molecule graphs with graph encoders and fine-tune the LLM to predict drug interactions. Other works [11, 18, 36, 44] explore how to solve graph learning tasks like node classification and graph classification by convert the graph structure information into LLMs.

Our research is relative to this topic as KGs also have complex graph structures on top of the text descriptions. In this paper, we will explore how to incorporate complex structural information in the KGs into the LLMs to achieve better reasoning capabilities on knowledge graph completion.

## 3 BASIC SETTINGS FOR LLM-BASED KGC

### 3.1 Notations and Preliminaries

A KG can be denoted as $\mathcal{G} = (\mathcal{E}, \mathcal{R}, \mathcal{T}, \mathcal{D})$ where $\mathcal{E}, \mathcal{R}$ are the entity set, relation set respectively. $\mathcal{T} = \{(h, r, t) \mid h, t \in \mathcal{E}, r \in \mathcal{R}\}$ is the triple set and $\mathcal{D}$ is the description set of each entity and relation. We denote $\mathcal{D}(e), \mathcal{D}(r)$ as the short textual description of each entity $e \in \mathcal{E}$ and each relation $r \in \mathcal{R}$. For example, the text description of the entity '/m/0ctzf1' is $\mathcal{D}$('/m/0ctzf1')='The Transformers'. When applying LLMs to KGC tasks, we denote a LLM as $\mathcal{M}$ that serves as a text decoder. The input textual sequence $\mathcal{S}$ of the model $\mathcal{M}$ consists of several parts: the instruction prompt $\mathcal{I}$, the triple prompt $\mathcal{X}$, and the optional auxiliary demonstration prompt $\mathcal{U}$. The instruction prompt $\mathcal{I}$ is the manually prepared instruction to guide the LLM $\mathcal{M}$ to execute the KGC task. The triple prompt $\mathcal{X}$ contains the textual information about the triples that need to be processed, which can be denoted as $\mathcal{X}(h, r, t) = \mathcal{D}(h) \oplus \mathcal{D}(r) \oplus \mathcal{D}(t)$, where $(h, r, t) \in \mathcal{T}$ is a triple and $\oplus$ denotes the textual token concatenation operation. In other words, the short descriptions of $h, r, t$ would be applied as the input information. The auxiliary demonstration prompt $\mathcal{U}$ is an optional prompt for different settings. In the following, we will follow this set of notations.

Meanwhile, we use triple classification as an entry point to investigate how to utilize LLM to accomplish the KGC task. Triple classification is a basic KGC task aiming to conduct binary classification tasks on the given triples. Whereas in the LLM paradigm, all tasks are converted into the form of text generation. Therefore, we desire the model $\mathcal{M}$ to answer true or false given the textual sequence input $\mathcal{S} = \mathcal{I} \oplus \mathcal{U} \oplus \mathcal{X}$.

Triple classification is different from vanilla text classification because the entities and the relations in the prompt have complex semantic information defined by the given KG. Without knowledge of this type of information, the model response is unreliable and unstable. Despite the vast amount of commonsense knowledge that exists in the LLMs [48], research has shown that large models are numb to fine-grained factual knowledge and will fall into a hallucination. Thus, incorporating the KG information into the prompt to provide more auxiliary information and guide the LLM to make structure-aware reasoning is the key to achieving excellent LLM-based KGC.

### 3.2 Extending Existing LLM Paradigms

In this section, we first discuss how to solve the KGC task with existing mainstream LLM paradigms called training-free reasoning approaches and instruction-tuning approaches.

*3.2.1 **Training-free reasoning approaches**.* Training-free reasoning approaches prompt the LLMs to get direct answers without training. Common training-free methods consist of zero-shot reasoning (ZSR) and in-context learning (ICL). For ZSR, we directly utilize the sequence $\mathcal{S}_{zsr} = \mathcal{I} \oplus \mathcal{X}$ as the input to get the prediction

results. The decoding process of the LLM $\mathcal{M}$ can be formulated as:

$$\begin{aligned} \mathcal{A}_{zsr} &= \arg\max_{\mathcal{A}} P_{\mathcal{M}}(\mathcal{A}|\mathcal{S}_{zsr}) \\ &= \arg\max_{\mathcal{A}} P_{\mathcal{M}}(\mathcal{A}|\mathcal{I}_{zsr}, \mathcal{X}) \end{aligned} \quad (1)$$

where $\mathcal{A}$ is the generated answer of the model $\mathcal{M}$ and $\mathcal{I}_{zsr}$ is the instruction template for ZSR. In the setting of ZSR, no KG information is added to the input sequence $\mathcal{S}_{zsr}$. The determinative information in the ZSR prompt is only the textual descriptions of the test triple. ZSR is unable to incorporate KG information due to its setting limitations, otherwise, it cannot be called zero-shot.

As another training-free paradigm, in-context learning (ICL) [9] allows the model $\mathcal{M}$ to add auxiliary demonstration $\mathcal{U}$ to the input $\mathcal{S}$ and accomplish the task in the form of analogical reasoning, which can be denoted as:

$$\begin{aligned} \mathcal{A}_{icl} &= \arg\max_{\mathcal{A}} P_{\mathcal{M}}(\mathcal{A}|\mathcal{S}_{icl}) \\ &= \arg\max_{\mathcal{A}} P_{\mathcal{M}}(\mathcal{A}|\mathcal{I}_{icl}, \mathcal{U}, \mathcal{X}) \end{aligned} \quad (2)$$

As for the triple classification task, the demonstration $\mathcal{U}$ should be some triples and their labels in the form of $\{(\mathcal{X}_i, y_i), 1 \le i \le k\}$, where $\mathcal{X}_i$ is the demonstration triple and $y_i$ is the label. We denote the ICL with $k$ demonstrations as $k$-shot ICL.

The demonstration triples can be randomly sampled from the existing training KG. However, to further incorporate the relative KG information of the test triple $(h, r, t)$, we propose to sample triples that are in the local structure of $h$ and $t$, which means one of the entities in each sampled triple should be $h$ or $t$. Besides, as existing KG only consists of positive triples, we employ negative sampling [21] to sample negative triples for demonstration. The number of positive and negative triples are the same for balanced predictions. In the demonstration prompt, the positive triples are labeled as true and the negative triples are labeled as false.

By doing this, we incorporate the local structural information into the demonstration prompt $\mathcal{U}$ with both positive and negative samples. Such a structure-aware demonstration could better enhance the analogical reasoning process of the model $\mathcal{M}$.

*3.2.2 **Instruction tuning approaches**.* Instruction tuning approaches fine-tune the LLMs with instruction template to activate the instruction following ability of LLMs. Vanilla instruction tuning leverages the input $\mathcal{S}_{it}$ to fine-tune LLMs. The instruction prompt $\mathcal{I}_{it}$ will describe the details of completing the triple classification task and the triple prompt $\mathcal{X}$ consists of the input triple. No other auxiliary demonstrations are included in the input template. To train the model $\mathcal{M}$, the input sequence is organized as $\mathcal{S}_{it} = \mathcal{I}_{it} \oplus \mathcal{X} \oplus \mathcal{A}_{it}$, where $\mathcal{A}_{it}$ is the predicted answer of the training data. The model $\mathcal{M}$ is fine-tuned with the next word prediction task [49] which is a universal approach to training LLMs. The training objective can be formulated as:

$$\mathcal{L}_{it} = -\frac{1}{|\mathcal{S}_{it}|} \sum_{i=1}^{|\mathcal{S}_{it}|} \log P_{\mathcal{M}}(s_i|s_{<i}) \quad (3)$$

where $s_i(i = 1, 2, \ldots, |\mathcal{S}_{it}|)$ represents the textual tokens of the input sequence $\mathcal{S}_{it}$. In the inference stage, the model $\mathcal{M}$ is employed to predict the answer $\mathcal{A}_{it}$ of the test data like Equation 1. Besides,

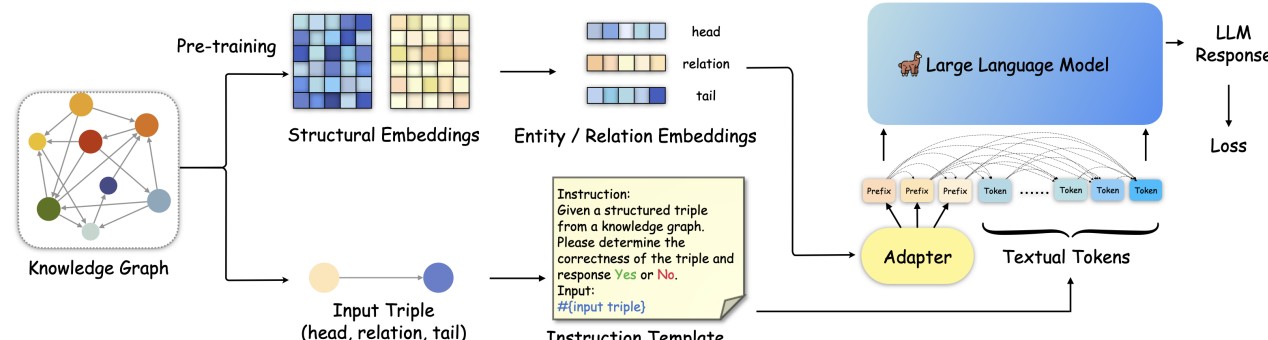

**Figure 2: An overview of the knowledge prefix adapter (KoPA) by us. KoPA first pre-trains structural embeddings for the entities and relations in the given KG and then employs instruction tuning to fine-tune the LLM. The structural embeddings of the given input triple will be projected into the textual space of the LLM by the adapter and serve as prefix tokens in the front of the input sequence, which can be "seen" by the following textual tokens due to the unidirectional attention mechanism in the decoder-only LLM.**

negative sampling [21] is also applied to generate negative data samples as training KG only consists of positve triples.

To incorporate semantic-rich KG information into LLMs, we also propose a structure-aware instruction tuning approach by adding the one-hop neighborhood structure information in the input prompt to inform the LLM with the local structural information. As mentioned before, the structural information of KG plays a significant role in the KGC tasks [37]. To incorporate such KG information during the fine-tuning stage, we achieve this goal by adding the neighborhood descriptions of the input triple. Specifically, we can sample the neighborhoods of the head $h$ and tail $t$ and put the textual descriptions of neighborhood triples in the demonstration prompt $\mathcal{U}_{it}$. In this way, the input training sequence is enhanced as $\mathcal{S}_{it} = \mathcal{I}_{it} \oplus \mathcal{U}_{it} \oplus \mathcal{X} \oplus \mathcal{A}_{it}$.

Therefore, we provide a detailed discussion of how the existing LLM paradigms can introduce local structural information about KGs to further enhance the model performance. However, though these approaches can work to some extent, they have obvious drawbacks. This  textbf fundamental approaches to incorporate KG structural information focus on **adding the neighborhood information to the input prompt in the text form**. However, representing the KG structural information in text is not a good choice, which may bring in more invalid or redundant information to the prompt. It's not scalable and effective to increase prompt length indefinitely because a long context will lead to both a decline in model capability and high computational consumption. Besides, we also have difficulty finding the structural information in the KGs that is decisive for triple discrimination. These two problems put us in a dilemma.

## 4 METHODLOGY

To solve such issues, we propose the **Kno**wledge **P**refix **A**dapter (**KoPA** for short) to incorporate the KG structural information into LLM for KGC. Figure 2 presents an intuitive view of KoPA. Firstly we extract the structural information of entities and relations from the KG through structural embedding pre-training, and then we

inform this structural information to LLM through a structural prefix adapter into the input sequence $\mathcal{S}$. The LLM $\mathcal{M}$ is further fine-tuned with the structural-enhanced text sequence. We will discuss the details in the next few sections about our design.

### 4.1 Structural Embedding Pre-training

Instead of adding text about the neighborhood information into the input sequence, KoPA extracts the structural information of the entities and relations by self-supervised structural embedding pre-training. For each entity $e \in \mathcal{E}$ and each relation $r \in \mathcal{R}$, we learn a structural embedding $e \in \mathbf{R}^{d_e}, r \in \mathbf{R}^{d_r}$ respectively, where $d_e, d_r$ are the embedding dimensions. We encode the KG structural information in the embeddings and further adapt them into the textual representation space of LLMs. Referring to the existing embedding-based KGC paradigm, we define a score function $\mathcal{F}(h, r, t)$ to measure the plausibility of the triple $(h, r, t)$. We adopt the self-supervised pre-training objective by negative sampling [3]:

$$
\begin{aligned}
\mathcal{L}_{pre} = \frac{1}{|\mathcal{T}|} \sum_{(h,r,t) \in \mathcal{T}} \Big( &- \log \sigma(\gamma - \mathcal{F}(h, r, t)) \\
&- \sum_{i=1}^{K} p_i \log \sigma(\mathcal{F}(h'_i, r'_i, t'_i) - \gamma) \Big)
\end{aligned}
\tag{4}
$$

where $\gamma$ is the margin, $\sigma$ is the sigmoid activation function and $(h'_i, r'_i, t'_i)(i = 1, 2, \ldots, K)$ are $K$ negative samples [3] of $(h, r, t)$. The weight $p_i$ is the self-adversarial weights proposed in [31].

By minimizing such a pre-training loss, the structural embeddings of each entity and relation are optimized to fit all its relative triples thus the KG structural information such as subgraph structure and relational patterns is captured in the embeddings. Such an approach has been proven effective in many embedding-based KGC methods [3, 31] to capture classic structural information like relational patterns and distributed entity representations [13] in the earliest days.

**Table 1: Comparasion among LLM-based KGC methods in three ways. As for the prompt length anaysis, $L_I, L_T$ denote the length of the instruction prompt and triple prompt. $L_D$ denotes the length of a demonstration and $k$ is the demonstration number. ZSR/ICL/IT refer to zero-shot reasoning, in-context learning, and instruction tuning respectively.**

| Method | Requires Fine-tuning | Extra KG Info | Prompt Length |
|---|---|---|---|
| ZSR | ✗ | ✗ | $L_I + L_T$ |
| ICL | ✗ | ✓ | $L_I + L_T + kL_D$ |
| Vanilla IT | ✓ | ✗ | $L_I + L_T$ |
| Enhanced IT | ✓ | ✓ | $L_I + L_T + kL_D$ |
| KoPA | ✓ | ✓ | $L_I + L_T + 3$ |

## 4.2 Knowledge Prefix Adapter

After structural embedding pre-training, we could obtain the structural embeddings $(h, r, t)$ of a triple $(h, r, t)$ where the KG structural information is encoded in. However, the structural embeddings are learned in a different representation space against the textual token representation space of the LLM $\mathcal{M}$, which means $\mathcal{M}$ can not directly understand these embeddings. Thus we apply a knowledge prefix adapter $\mathcal{P}$ to project them into the textual token representation space of $\mathcal{M}$. Specifically speaking, the structural embeddings are converted to several virtual knowledge tokens $\mathcal{K}$ by $\mathcal{P}$:

$$\mathcal{K} = \mathcal{P}(h) \oplus \mathcal{P}(r) \oplus \mathcal{P}(t) \tag{5}$$

In practice, the adapter $\mathcal{P}$ would be a simple projection layer [50]. Then we put $\mathcal{K}$ in the front of the original input sequence $S$ serving as a prefix of the instruction and triple prompt $S_{kpa} = \mathcal{K} \oplus \mathcal{I}_{it} \oplus \mathcal{X}$. This way, all the following text tokens can be seen with the prefix $\mathcal{K}$ due to the unidirectional attention in decoder-only LLMs. By doing this, the textual tokens can pay unidirectional attention to the structural embeddings of the input triple. Such a structure-aware prompt will be employed during fine-tuning and inference. During training, we froze the pre-trained structural embeddings. The adapter is optimized to learn the mapping from structural knowledge toward textual representation and will have the generalization to new triples in the inference stage, which will benefit the textual description and provide the triple information from another perspective to make enhanced predictions.

## 4.3 Complexity Analysis

After proposing KoPA, we make a comparison among LLM-based KGC methods to demonstrate the advantages of KoPA, which is shown in Table 1. Compared with the basic paradigms (ZSR/ICL/IT), KoPA incorporates the KG structural embeddings into LLM to combine the textual and structural information. Meanwhile, KoPA makes the length of the prompt more refined as the length of virtual tokens generated by the structural prefix adapter is fixed to 3 for head/relation/tail respectively. In contrast, the prompt length of structure-aware IT (enhanced IT in the table) is linearly related to the number of neighborhood triples $k$. In contrast to methods that incorporate structural information based on textual descriptions, KoPA achieves this goal by fixed-length virtual knowledge tokens generated by the adapter.

**Table 2: Statistical information of datasets. The positve (+) and negative (-) samples are 1:1 in the valid / test set.**

| Dataset | $|\mathcal{E}|$ | $|\mathcal{R}|$ | #Train | #Valid(+/-) | #Test(+/-) |
|---|---|---|---|---|---|
| UMLS | 135 | 46 | 5216 | 652/652 | 661/661 |
| CoDeX-S | 2034 | 42 | 32888 | 1827/1827 | 1828/1828 |
| FB15K-237N | 13104 | 93 | 87282 | 7041/7041 | 8226/8226 |

## 5 EXPERIMENTS

### 5.1 Experimental Settings

*5.1.1* **Datasets**. In our experiments, we use three public KG benchmarks UMLS [40], CoDeX-S [28], and FB15K-237N [21] to evaluate the proposed LLM-based KGC methods. The detailed split information of the datasets is shown in Table 2.

*5.1.2* **Baseline Methods**. In our experiments, we provide a comprehensive comparison with three broad classes of baselines on triple classification, which is an important subtask of KGC. The KGC baselines can be divided into three parts: embedding-based methods [3, 31, 34, 38], PLM-based methods [21, 40], and LLM-based methods [41]. Besides, we further divide the LLM-based methods into two categories: training-free methods and fine-tuning methods. Training-free methods consist of ZSR and ICL while fine-tuning methods consist of vanilla IT and structure-aware IT (enhanced IT). The specific models used for these baselines are listed below:

(1). **Embedding-based KGC methods**. We select four traditional embedding-based KGC methods for comparisons, namely TransE [], DistMult [38], ComplEx [34], and RotatE [31]. These methods predict the triple plausibility by the learned structural embeddings and the score functions defined in the model.

(2). **PLM-based KGC methods**. We select KG-BERT [40] and PKGC [21] as PLM-based KGC baselines, which are classic methods focusing on the triple classification task. These methods treat triple classification as a binary text classification task.

(3). **LLM-based KGC methods**. LLM-based KGC research is still at an early stage. There are only KGLLaMA [41] to be the LLM-based KGC baseline. In addition to KGLLaMA, the methods proposed in Section 3 by us including ZSR, ICL, IT, and structure-aware IT (enhanced IT) will also serve as baselines.

*5.1.3* **Implementation and Detail Settings**. We reproduce the baseline results and implement the KoPA proposed by us.

For embedding-based KGC methods, we reproduce the results with OpenKE we set the embedding dimension $d_e = d_r = 512$ and sample $K = 32$ negative samples during training. The margin $\gamma$ is tuned among $\{0, 4, 6, 8, 12\}$. After training KGC models, we search for the best classification score threshold on the validation set for test data following the traditional setting [3].

For PLM-based methods, the backbone model for PLM-based KGC methods is BERT [8]. We fine-tune the KG-BERT according to the official code implementation. Since PKGC requires a lot of manual work to annotate each relation with a prompt, we only report the results of FB15K-237N shown in the original paper.

For zero-shot reasoning, in addition to measuring with the same backbone Alpaca, we also test the performance of the *GPT-3.5-turbor*

**Table 3: The main experiment results of triple classification. We report the accuracy (ACC), precision (P), recall (R), and F1-score (F1) results for each method on the three datasets. "-" means the result are missing because the specificity of PKGC makes it difficult to reproduce. The best Acc / F1 results in baselines are marked with underline, and we highlight our results with bold when we achieve new SOTA.**

| | Model | UMLS | | | | CoDeX-S | | | | FB15K-237N | | | |
|---|---|---|---|---|---|---|---|---|---|---|---|---|---|
| | | Acc | P | R | F1 | Acc | P | R | F1 | Acc | P | R | F1 |
| Embedding-based | TransE [3] | 84.49 | 86.53 | 81.69 | 84.04 | 72.07 | 71.91 | 72.42 | 72.17 | 69.71 | 70.80 | 67.11 | 68.91 |
| | DistMult [38] | 86.38 | 87.06 | 86.53 | 86.79 | 66.79 | 69.67 | 59.46 | 64.16 | 58.66 | 58.98 | 56.84 | 57.90 |
| | ComplEx [34] | 90.77 | 89.92 | 91.83 | 90.87 | 67.64 | 67.84 | 67.06 | 67.45 | 65.70 | 66.46 | 63.38 | 64.88 |
| | RotatE [31] | 92.05 | 90.17 | 94.41 | 92.23 | 75.68 | 75.66 | 75.71 | 75.69 | 68.46 | 69.24 | 66.41 | 67.80 |
| PLM-based | KG-BERT [40] | 77.30 | 70.96 | 92.43 | 80.28 | 77.30 | 70.96 | 92.43 | 80.28 | 56.02 | 53.47 | 97.62 | 67.84 |
| | PKGC [21] | - | - | - | - | - | - | - | - | 79.60 | - | - | 79.50 |
| LLM-based Training-free | Zero-shot(Alpaca) | 52.64 | 51.55 | 87.69 | 64.91 | 50.62 | 50.31 | 99.83 | 66.91 | 56.06 | 53.32 | 97.37 | 68.91 |
| | Zero-shot(GPT-3.5) | 67.58 | 88.04 | 40.71 | 55.67 | 54.68 | 69.13 | 16.94 | 27.21 | 60.15 | 86.62 | 24.01 | 37.59 |
| | ICL(1-shot) | 50.37 | 50.25 | 75.34 | 60.29 | 49.86 | 49.86 | 50.59 | 50.17 | 54.54 | 53.67 | 66.35 | 59.34 |
| | ICL(2-shot) | 53.78 | 52.47 | 80.18 | 63.43 | 52.95 | 51.54 | 98.85 | 67.75 | 57.81 | 56.22 | 70.56 | 62.58 |
| | ICL(4-shot) | 53.18 | 52.26 | 73.22 | 60.99 | 51.14 | 50.58 | 99.83 | 67.14 | 59.29 | 57.49 | 71.37 | 63.68 |
| | ICL(8-shot) | 55.52 | 55.85 | 52.65 | 54.21 | 50.62 | 50.31 | 99.83 | 66.91 | 59.23 | 57.23 | 73.02 | 64.17 |
| LLM-based Fine-tuning | KG-LLaMA [41] | 85.77 | 87.84 | 83.05 | 85.38 | 79.43 | 78.67 | 80.74 | 79.69 | 74.81 | 67.37 | 96.23 | 79.25 |
| | KG-Alpaca [41] | 86.01 | 94.91 | 76.10 | 84.46 | 80.25 | 79.38 | 81.73 | 80.54 | 69.91 | 62.71 | 98.28 | 76.56 |
| | Vanilla IT | 86.91 | 95.18 | 77.76 | 85.59 | 81.18 | 77.01 | 88.89 | 82.52 | 73.50 | 65.87 | 97.53 | 78.63 |
| | Structure-aware IT | 89.93 | 93.27 | 86.08 | 89.54 | 81.27 | 77.14 | 88.40 | 82.58 | 76.42 | 69.56 | 93.95 | 79.94 |
| KoPA | | **92.58** | 90.85 | 94.70 | **92.70** | **82.74** | 77.91 | 91.41 | **84.11** | 77.65 | 70.81 | 94.09 | **80.81** |

which has 175B parameters. For the in-context learning method, we sample k-shot (k=1,2,4,8) structure-aware demonstrations. Besides, we sample 4 neighborhood triples for each triple to conduct structure-aware instruction tuning. For KoPA, we employ RotatE [31] and the score function of structural embedding pre-training and the embedding dimension is set to 512 and the adapter is a 512×4096 linear projection layer.

For KoPA, we employ Alpaca-7B [32] as the LLM backbone. Alpaca is a famous extended version of LLaMA [33] model fine-tuned on instruction-following data. We reproduce the triple classification results of KGLLaMA [41] over two backbones (LLaMA and Alpaca) to avoid the effect of backbone choice on the results. We name the two baseline models KGLLaMA and KGAlpaca respectively. For all the fine-tuning methods (instruction tuning, structure-aware instruction tuning, and KoPA), we fine-tune Alpaca using LoRA [14] with rank 64. The number of epochs is searched in {3, 4, 5} and the learning rate is tuned in $\{1e^{-4}, 3e^{-4}, 5e^{-4}\}$. We use the AdamW optimizer [20] with a fixed batch size of 12. We conducted all the experiments with Nvidia A800 GPUs. The structural embedding pre-training process is efficient and only takes several minutes to finish. Therefore, the main time cost is caused by the LLM fine-tuning, which takes several hours for different datasets. (1 hour for UMLS, 4 hours for CoDeX-S, and 8 hours for FB15K-237N in our experimental environments).

*5.1.4* **Evaluation Protocol**. We evaluate the methods with the triple classification task [3], which is essentially binary classification and all the test datasets are label-balanced. Therefore, we use accuracy, precision, recall, and F1-score as the evaluation metrics.

## 5.2 Main Results

The main experiment results of triple classification are shown in Table 3. Since precision and recall alone do not give a good response to the model's performance on the classification task, we focus on accuracy and F1-score. However, to provide a comprehensive analysis of different models, we also report the precision and recall results in the table. Overall, we can find that KoPA achieves outperforming accuracy and F1 results compared with the existing 16 baseline models on all three datasets. Taking CoDeX-S as an example, KoPA achieves 1.81% improvement in accuracy and 1.85% improvement on F1. As we use the pre-trained RotatE embeddings in KoPA, we can observe that KoPA significantly outperforms the original embedding-based RotatE method, especially on larger and more challenging datasets like CoDeX-S and FB15K-237N.

Meanwhile, compared with all LLM-based approaches, we can see that the LLMs cannot understand the KG structural information well without fine-tuning. The zero-shot LLMs perform very poorly in the triple classification task even though GPT-3.5-turbo (175B parameters) has excellent capability. Though the demonstrations provided by ICL can incorporate the KG information, the performance gain is limited. Besides, the prediction results of training-free methods are biased and easy to slip into the extremes of all-right or all-wrong, as the recall of them is either very high or very low but the F1 scores are relatively low all the time.

However, fine-tuning LLMs can introduce the KG information into LLMs as the overall performance makes obvious improvements. Meanwhile, though structure-aware IT enhances the input prompt with neighborhood information of triples, its performance is also

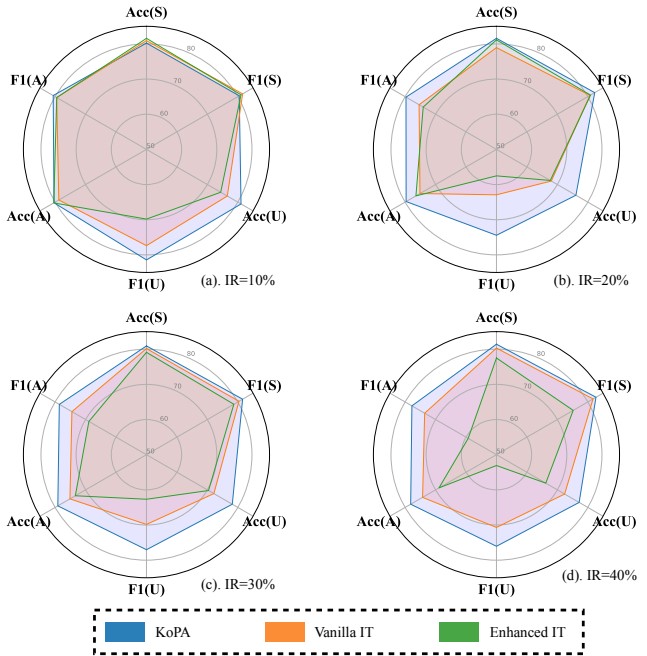

**Figure 3: The results of the transferbility experiment. We report the results on CoDeX-S dataset under different inductive rate (IR). Besides, we split the test data into seen (S) and unseen (U) parts based on whether the entity appeared during training. Also we total the results of all (A) the test data together. Accuracy (Acc) and F1-score (F1) are reported in the radar charts.**

limited compared with KoPA. This suggests that the structural embeddings consist of more semantic-rich information compared with text-based auxiliary prompts, which can also be understood by the LLM through the prefix adapter. Combining the analysis in Section 4.3 and the experimental results, KoPA achieves better results on top of shorter prompts.

## 5.3 Transferability Exploration

The results in the main experiments have shown the effectiveness of KoPA. To further validate the generality and the transferability of KoPA, we conduct a new transferability experiment. In this experiment, we will demonstrate that the knowledge prefix adapter will learn to transfer from structural embeddings to textual token representations and provide semantic-rich auxiliary information to enhance the decoding process of LLM inference.

We demonstrate this point by testing the influence of KoPA for entities that do not appear in the training phase, which is also called inductive setting in other KGC works [6]. We split the KG dataset into an inductive setting with a defined inductive rate (IR), which refers to the ratio of unseen entities during training. For example, if IR=10%, we will randomly select 10% entities as the inductive entity set. Any triple in the training set whose head or tail is in the inductive set will be removed during training. Besides, the triples in the test set will be divided into two parts: the seen (S) part and the unseen (U) part. If the head or tail in a triple is in the inductive entity set, it will be regarded as unseen. We fine-tune the LLM with

| Model | | Acc | F1 |
|---|---|---|---|
| KoPA(Prefix + RotatE) | | 82.74 | 84.11 |
| Embedding | w/o SE | 81.18 | 82.52 |
| | w/ TransE | 82.46 | 83.42 |
| | w/ DistMult | 80.71 | 81.27 |
| | w/ ComplEx | 81.21 | 82.12 |
| | w/ Random | 81.53 | 82.36 |
| Position | Infix | 81.21 | 82.69 |
| | Suffix | 77.29 | 77.75 |

only remaining seen triples and test on both seen and unseen triples. In this setting, a set of entities will not participate in the training process and the LLM does not see their textual descriptions, which will make the test process more challenging. We report the accuracy and F1 score for seen (S), unseen (U), and all (A) test triples, which is shown in Figure 3 for three fine-tuning methods: KoPA, vanilla IT, and structure-aware IT (enhanced IT in the figure).

From the radio charts, we can observe that KoPA outperforms the other methods for unseen triples and has less performance degradation when IR increases. The performance of structure-aware IT (enhanced IT) with neighborhood triples in the textual form is more unstable. These phenomena suggest that the knowledge prefix adapter can learn a good mapping from the structural embeddings to the textual representation, which is transferable even if the entities are unseen during training. The structural embeddings captured from KG play a more significant role in informing the LLM with useful structural information.

## 5.4 Ablation Study

To verify the effectiveness of the KoPA design, we conduct a two-part ablation study. The first part is designed to verify the effectiveness of structural embedding and the second part is designed to verify the effectiveness of prefix adapter. As shown in Table 4, we can find that removing the structural embeddings or replacing them with random initialized embeddings both lead to performance decline. Also, we find that the model is compatible with different types of structural embeddings. However, the performance gain depends on whether the embedding was originally powerful in the triple classification task or not. Refer to Tables 3, TransE [3] and RotatE [31] are better embedding-based KGC models compared with DistMult [38] and ComplEx [34]. This demonstrates that semantic-rich structural information is the key to performance improvement and KoPA takes full advantage of it.

Meanwhile, putting the virtual knowledge tokens generated by the adapter in the middle (infix) or in the last (suffix) of the input sequence will also decrease the performance. We believe the reason is that putting tokens in the front of the sequence will make all the text pay attention to them as LLMs are usually decoder-only architectures with unidirectional self-attention. Then the LLM can make a better decision with the structural embeddings that fully interact with the text. Combining these two parts of the ablation study, we believe that our design of KoPA is effective and reasonable.

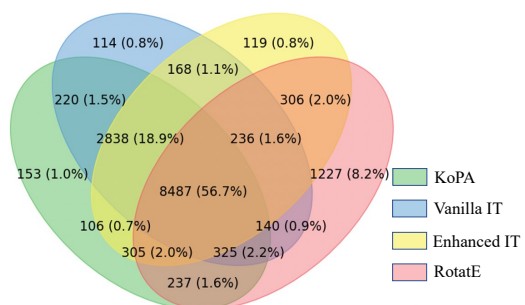

Figure 4: The Venn diagram of the correct predictions from various KGC models. Each intersecting part in the diagram represents the same predictions from different models on certain data.

## 5.5 Case Study

To make a more intuitive view of KoPA, we conduct a case study in this section from both macro and micro perspectives. From a macro perspective, we count the prediction overlap of several models and plot a Venn diagram shown in Figure 4.

From the diagram we can find that KoPA has a significant portion of the proper predictions that do not intersect with several other models, which means that KoPA makes the right prediction on some test data that many other models predict incorrectly. This suggests that the structural information incorporated in KoPA has a significant role in making correct predictions. For a micro example, a test triple (*John Landis, film director film, Coming to America*) is predicted as wrong by the RotatE model and vanilla instruction tuning LLM. With retrieved neighborhood triples (*Coming to America, locations, New York City*), (*John Landis, nationality, USA*), (*Coming to America, genre, romantic comedy*), (*Comedy, common netflix titles, Coming to America*), the structure-aware fine-tuned LLM still makes a wrong prediction because the neighborhood information is of little use in the judgment of the current prediction though they are the correct factual. The structural embeddings applied in KoPA contain more information than structural information in the form of text and are easier for us to extract by a structural pre-training process. Thus, KoPA outperforms other models in the triple classification task.

## 5.6 Common Ability Retention

To delve into the preservation of generic capabilities in LLMs, we conducted another experiment to assess the overall proficiency of LLMs both before and after fine-tuning. We apply the MMLU [12] benchmark for this problem. MMLU is the most popular benchmark to evaluate the general abilities of LLMs in different domains such as Humanities, Social Sciences, STEM, and others. The overall evaluation results on different datasets are shown in Figure 5:

From the results, it can be noticed that after KoPA training, there were discernible alterations in the generalized abilities of LLMs. In most instances, there was a decrease, but notably, STEM proficiency exhibited improvement on the UMLS dataset. We attribute this phenomenon to the UMLS being a medical KG, encompassing substantial knowledge in medicine, biology, and chemistry, and training on this dataset allows the model to acquire more STEM knowledge. Consequently, when facing natural language inputs

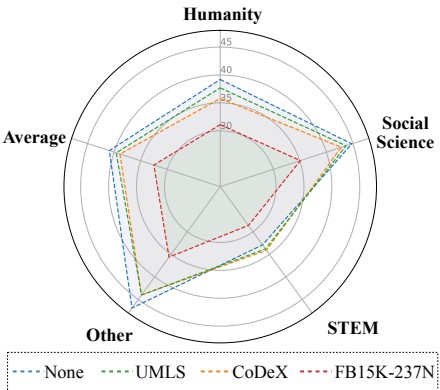

Figure 5: The common ability experiments on MMLU.

differing from the training task, the model adeptly leverages the acquired knowledge from KGC task fine-tuning to get enhanced results. We have listed several subjects in MMLUs that showed improvement after training with UMLS. These subjects are highly relevant and close to the knowledge domain encapsulated in the UMLS in Table 5. The LLMs trained with the KGC task also achieved significant improvements across different input prompts, marking a compelling observation.

Table 5: The specific domains in MMLU in which LLM achieves higher scores after training on UMLS.

| Subjects | w/o Training | w/ Training |
|---|---|---|
| Clinical | 44.9 | **47.9** (+3.0%) |
| College Medicine | 30.1 | **31.2** (+1.1%) |
| High School Biology | 42.9 | **46.8** (+3.9%) |
| High School Chemistry | 30.0 | **32.0** (+2.0%) |
| Medical Genetics | 44.0 | **48.0** (+4.0%) |

## 6 CONCLUSION

In this paper, we systematically explore how to incorporate structural understanding ability into LLMs to make structure-aware reasoning for KGC tasks. We extend the original LLM paradigms and propose structure-aware ICL and IT methods to incorporate the structural information by text. We further propose KoPA, a knowledge prefix adapter to incorporate the pre-trained structural embeddings into the LLMs. We conduct triple classification experiments to make comprehensive comparisons among the structure-aware methods and demonstrate the outperforming results achieved by KoPA. In the future, we plan to dive deep into LLM-based KGC and think about a more unified framework to accomplish all the KGC tasks with LLMs. Besides, we will also explore flexibly adapting KGs into LLM-based downstream applications to make the LLMs knowledgeable, reliable, and human-friendly.

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
