# OpenReview forum: "Making Large Language Models Perform Better in Knowledge Graph Completion"
_acmmm.org/ACMMM/2024/Conference — MM2024 Oral_

### Official Review · Reviewer_8M6Q · 2024-05-15

**Rating:** 5
**Confidence:** 3

**Summary:**

The paper presents a framework for knowledge graph completion that utilizes large language models and prefix tuning to incorporate both structural and semantic information. The authors conduct extensive experiments on multiple datasets, demonstrating that their proposed method achieves state-of-the-art performance. The ablation study and detailed analysis further support the effectiveness of the framework.

**Strengths:**

1. The motivation behind the research is clear and the proposed method is sound.
2. The paper includes abundant experiments with excellent results, showcasing the effectiveness of the proposed approach.
3. The writing is clear and the paper is easy to follow, aided by well-designed figures.

**Limitations:**

There are no significant disadvantages in this paper.

This paper mainly focuses on text-attributed knowledge graphs and large language models. In this situation, the reference to ``Multimedia'' topic seems somewhat far-fetched.

**Suitability:**

2

---

### Official Review · Reviewer_PiDQ · 2024-05-23

**Rating:** 5
**Confidence:** 3

**Summary:**

This paper proposes a novel framework knowledge prefix adapter (KoPA) to combine structural knowledge graphs (KG) and large language models (LLMs) together to achieve cross-modal structure-aware reasoning in the textual representation space. KoPA first pre-trains on the knowledge graphs and inject the structural knowledge into LLMs with a prefix adapter.

**Strengths:**

- This paper focuses on the combination of KGs and LLMs, which is a very important topic nowadays. The authors have devised a concise and effective approach to cross-modal knowledge transfer using the prefix adapter method.
- The workload of the experiment is sufficient, not only for KGC but also for more in-depth investigations such as experiments on the transferability of methods.
- I think the final test on the common ability retention is very interesting. It explores the extent to which the LLM retained knowledge in generalized domains after cross-modal knowledge injection. The conclusions obtained from these experiments are interesting and instructive.

**Limitations:**

- I have concerns about the experiment settings. The author simplifies the KGC problem as a binary classification, which is not very convincing. Existing studies in the KGC usually use ranking metrics like Hit@k, and recall. The binary classification is an over-simplified KGC setting, which might not be practical in reality.
- For structural-aware instruction tuning in the base LLMs paradigms proposed as baselines in the paper, some more detailed examples are missing to show how structural information is injected into the text.
- There are some typos in the paper. For example, GPT-3.5-turbor --> GPT-3.5-turbo in line 579. Missing reference for TransE at Sec 5.1.2

**Suitability:**

3

---

### Official Review · Reviewer_m97p · 2024-05-24

**Rating:** 4
**Confidence:** 4

**Summary:**

The paper proposes a novel KoPA framework, which incorporates the pre-trained structural embeddings and knowledge prefix adapter into the LLMs, thereby facilitating LLM’s factual knowledge reasoning ability. Experiments and analysis demonstrate the outperforming results for KGC tasks. Furthermore, the paper evaluates the proposed KoPA's generic capability preservation via the MMLU benchmark to make LLMs knowledgeable, reliable, and human-friendly.

**Strengths:**

1. The paper designs a novel KoPA, which effectively integrates pre-trained KG structural embeddings with LLMs and aligns cross-modal interactions between textual embeddings from LLMs and structural embeddings sourced from KGs.
2. The paper clearly and detailedly introduces the existing LLM paradigms (i.e., training-free reasoning and instruction tuning approaches) and elaborates on their shortcomings.
3. The paper conducts comprehensive experiments to evaluate the proposed KoPA’s factual knowledge reasoning ability, transferability, and common ability retention.
4. The paper is well-written and can be easy to follow. Moreover, the proposed solution is clear and technically sound.

**Limitations:**

1. The adopted negative sample strategy and score function $\mathcal{F}$ in Section 4.1 need to be further described. In Figure 1, we can observe that compared with useless structural information, useful structural information facilitates LLM for KGC tasks. Hence, I wonder if you design some negative sample strategies to learn structural embeddings.
2. The efficiency analysis of KoPA should be discussed. Although KoPA improves the accuracy of KGC tasks, the training time and inference time of LLMs still incur a very high computational cost and time.
3. Please explain the necessity of four metrics (i.e., Accuracy, Precision, Recall, and F1-score) instead of MRR and Hit@K which are commonly used in these works for KGC [1-4].
4. There are some typos in the article. For example, in Line 552, "TransE [], DistMult" should be revised as "TransE [3], DistMult".

**Reference**
[1] KC-GenRe: A Knowledge-constrained Generative Re-ranking Method Based on Large Language Models for Knowledge Graph Completion, LREC-COLING, 2024.
[2] Multi-perspective Improvement of Knowledge Graph Completion with Large Language Models, LREC-COLING, 2024.
[3] KICGPT: Large Language Model with Knowledge in Context for Knowledge Graph Completion, EMNLP, 2023.
[4] Contextualization Distillation from Large Language Model for Knowledge Graph Completion, EACL, 2024.

**Suitability:**

3

---

### Meta-Review · Area_Chair_Fwkc · 2024-07-03

**Recommendation:** Accept (Oral)
**Confidence:** 4

**Metareview:**

The paper presents a novel approach, KoPA, integrating pre-trained KG structural embeddings with LLMs, enhancing cross-modal interactions for KGC tasks. Reviewers have praised the comprehensive experimental validation, clear writing, and sound methodology. However, concerns were raised regarding the detailed description of the negative sample strategy, the efficiency analysis, and the oversimplification of the KGC problem as a binary classification. Additionally, the necessity of four metrics over conventional ones like MRR and Hit@K, as well as the practical applicability of the approach in multimedia contexts, were questioned.